# Development of High-Performance Flexible Radiative Cooling Film Using PDMS/TiO_2_ Microparticles

**DOI:** 10.3390/mi14122223

**Published:** 2023-12-10

**Authors:** Junbo Jung, Siwon Yoon, Bumjoo Kim, Joong Bae Kim

**Affiliations:** 1Department of Mechanical Engineering, Kongju National University, Cheonan 31080, Republic of Korea; junjoung3167@smail.kongju.ac.kr; 2Energy Efficiency Research Division, Korea Institute of Energy Research, Daejeon 34129, Republic of Korea; siwon@kier.re.kr; 3Department of Mechanical and Automotive Engineering, Kongju National University, Cheonan 31080, Republic of Korea; 4Department of Future Convergence Engineering, Kongju National University, Cheonan 31080, Republic of Korea

**Keywords:** radiative cooling, microparticles, flexibility, zero-energy, thermal management

## Abstract

Radiative cooling, which cools an object below its surrounding temperature without any energy consumption, is one of the most promising techniques for zero-energy systems. In principle, the radiative cooling technique reflects incident solar energy and emits its thermal radiation energy into outer space. To achieve maximized cooling performance, it is crucial to attain high spectral reflectance in the solar spectrum (0.3–2.5 μm) and high spectral emittance in the atmospheric window (8–13 μm). Despite the development of various radiative cooling techniques such as photonic crystals and metamaterials, applying the cooling technology in practical applications remains challenging due to its low flexibility and complicated manufacturing processes. Here, we develop a high-performance radiative cooling film using PDMS/TiO${}_{2}$ microparticles. Specifically, the design parameters such as microparticle diameter, microparticle volume fraction, and film thickness are considered through optical analysis. Additionally, we propose a novel fabrication process using low viscosity silicone oil for practical fabrication. The fabricated film accomplishes 67.1 W/m${}^{2}$ of cooling power, and we also analyze the cooling performance difference depending on the fabrication process based on the measurement and optical calculation results.

## 1. Introduction

Recently, there has been a growing emphasis on eco-friendly technologies to address global environmental challenges, including issues such as global warming, abnormal climate patterns, and ecosystem changes [1]. Radiative cooling, which enables the cooling of an object below the ambient temperature without any energy consumption, has drawn significant attention [2,3,4,5]. In principle, it is crucial to create an ideal surface with high spectral reflectance in the solar spectrum (0.3–2.5 μm) and high spectral emittance in the atmospheric window (8–13 μm) for radiative cooling [6,7,8].

To demonstrate the ideal surface of radiative cooling, various approaches including multilayers, metamaterials, randomly distributed particle structures, and porous structures [2,9,10,11,12,13,14,15,16,17] have been intensively investigated. Among these, photonic structures, including metamaterials [18,19,20], multilayers [21,22], and grating patterns [23,24,25], exhibit ideal optical characteristics such as high spectral reflectance in the solar spectrum and high spectral emittance in the mid-infrared regime. Nevertheless, their practical applications are impeded by challenges such as scalability, complex fabrication processes, and limited flexibility [7,13,26]. Flexibility, scalability, and simple fabrication process are crucial for applying radiative cooling techniques, but photonic structures still pose challenges in satisfying these requirements [7,27].

In contrast, particle-based structures [28,29,30,31,32], offering both flexibility and superior cooling performance, are promising candidates for a practical approach to radiative cooling. They are typically composed of microparticles for scattering in the solar spectrum and a medium for enhancing the thermal emission in an atmospheric window. Media such as polydimethylsiloxane (PDMS), polyvinyl chloride (PVC), or polymethyl methacrylate (PMMA) are commonly employed [33,34,35], with the addition of TiO${}_{2}$, Al${}_{2}$O${}_{3}$, and SiO${}_{2}$ microparticles to enhance the reflectance in the visible regime. Due to their simple compositions, the particle-based structures are easier to fabricate compared to photonic structures, ensuring excellent flexibility and scalability [36,37].

Despite the advantages of particle-based structures, they encounter significant challenges in practical applications [38]. The primary issues include the difficulty of achieving uniform dispersion of microparticles in a highly viscous medium, the risk of cracking caused by thermal shrinkage and expansion, and the use of toxic materials in the fabrication process [39,40]. When PMMA is employed as a medium for a particle-based structure, it poses challenges in terms of sustaining its performance for a long time on the exterior walls of buildings or vehicles due to its thermoplastic nature. Moreover, its inherent brittleness is inadequate for practical applications [41].

In addition, PVC and PDMS, exhibiting high viscosity, have issues in terms of the uniform dispersion of microparticles during the fabrication process [42]. Although there are conventional methods such as a vortex mixer, stirrers, or ultrasonicators to disperse microparticles in a fluid, it is almost impossible to disperse particles uniformly when the medium has an extremely high viscosity, as with PVC and PDMS. Although it can be partially mitigated by lowering the concentration of microparticles, it leads to an inevitable decrease in the cooling performance of films. These days, several solutions to these challenges have been proposed by utilizing volatile fluids such as toluene and acetone as a buffer medium [43,44]. However, employing toxic materials such as toluene and acetone raises concerns about the feasibility of the large-scale production and practical applications of radiative cooling. Hence, it is necessary to develop a practical fabrication process as well as a high-performance radiative cooling structure employing safe materials.

The present study proposes a novel fabrication process and structure of a flexible radiative cooling film including microparticles for obtaining enhanced cooling performance. Specifically, we determine the material’s composition, the microparticle diameter, microparticle volume fraction, and film thickness during the design process. This detailed design process enables us to anticipate the cooling performance before practical fabrication, offering valuable insights into the effectiveness of the proposed film. In the proposed fabrication process, we employ silicone oil, volatile and safe, as a buffer medium to tackle the dispersion issue and lack of flexibility arising from the high viscosity of PDMS. Consequently, this study introduces a practical fabrication process and experimentally demonstrates a flexible radiative cooling structure based on microparticles. The flexible radiative cooling film ensures not only a high cooling performance but also enhances productivity and safety.

## 2. Materials and Methods

### 2.1. Materials

Polydimethylsiloxane (PDMS) precursors, including PDMS elastomer base A and curing agent B were purchased from Dow Corning Co., Ltd. (Midland, MI, USA). The kinematic viscosity and density of PDMS elastomer base is 5000 cSt at 25 °C and 1.11 g/cm${}^{3}$ in the liquid state. The kinematic viscosity and relative density of curing agent is 110 cSt at 25 °C and 1.03 (water = 1). Titanium oxide (TiO${}_{2}$, rutile, >99.5%, 0.2–0.4 μm) was obtained from Ditto Technology Co., Ltd. (Gunpo, Republic of Korea), and density of TiO${}_{2}$ microparticle is 4.2 g/cm${}^{3}$. Silicone oil (SF1000N, 1 cSt) was obtained from KCC Silicone Corp. (Seoul, Republic of Korea) and has a specific gravity of 0.808.

### 2.2. Design

Prior to the fabrication of flexible radiative cooling (FRC) films, this study proposes the material composition and structure necessary to achieve enhanced cooling performance. According to previous research, the maximum cooling performance attainable through radiative cooling is 150 W/m${}^{2}$ [45]. To reach this maximum cooling performance, the ideal radiative cooling material should be designed by considering optical properties in the solar spectrum and the atmospheric windows. As shown in Figure 1a, when the incident solar irradiation is completely reflected, and the thermal radiation from the target is transferred entirely to space, the spontaneous cooling from radiative cooling films reaches its maximum performance.

Figure 1a simply illustrates the proposed design of this study and the mechanism of particle-based radiative cooling. As depicted in Figure 1a, we employ the particle-based radiative cooling approach, which consists of a polymer matrix and microparticles, to enhance flexibility and productivity. In detail, the embedded microparticles in the polymer matrix scatter the sunlight, and the polymer, with high spectral emittance, radiates the thermal energy of the target transferred through heat conduction. The polymer satisfying our design objectives is PDMS, which has excellent flexibility and high thermal emission. It exhibits higher absorptance (=emittance) in the mid-infrared regime compared to other polymer matrices such as polycarbonate (PC), polyethylene terephthalate (PET), polymethylmethacrylate (PMMA), and polyvinylidene fluoride (PVDF) [46]. Thus, we choose PDMS as the polymer matrix with it having both high flexibility and high spectral emittance characteristics at once.

There are several candidate materials for high scattering in the solar spectrum, such as SiO${}_{2}$, TiO${}_{2}$, and Al${}_{2}$O${}_{3}$. Since reflectance in the solar spectrum significantly affects cooling performance, the decision regarding the microparticle material and its diameter is a crucial and complex step. Therefore, we conducted an analysis of the optical properties (scattering and absorption efficiencies, $Q_{scat}$ and $Q_{abs}$, respectively) of each microparticle using Mie scattering theory. Mie scattering theory is an analytical solution derived from Maxwell’s equations to calculate the scattering and absorption efficiencies of spherical microparticles [15]. These efficiencies are computed using the intrinsic optical properties of each particle material and the surrounding medium, such as PDMS. Each efficiency derived from Mie scattering theory can be represented by the following infinite series [47,48]: (1)$$Q_{scat}=\frac{2}{\chi^{2}}\sum_{n=1}^{\infty }\left(2n+1\right)\left[\left({\left|a_{n}\right|}^{2}+{\left|b_{n}\right|}^{2}\right)\right]$$
(2)Qabs=2χ2∑n=1∞(2n+1)Re(an+bn)−Qscat
where size parameter χ is given by parameter $\pi Dn_{m}/\lambda$ with the diameter of spherical microparticle *D*, the refractive index of medium $n_{m}$, and the wavelength λ. Re is a real part of a complex number, and *n* is mode index. $a_{n}$ and $b_{n}$ represent the coefficient of the Mie scattering theory. The scattering and absorption efficiencies relate to how efficiently microparticles scatter or absorb the incident light.

Figure 1b depicts the scattering and absorption efficiencies based on the microparticle material. The solid and dotted lines represent the scattering and absorption efficiencies, respectively. In terms of scattering efficiency, SiO${}_{2}$ exhibits the smallest value among the candidates. The maximum scattering efficiency of the Al${}_{2}$O${}_{3}$ microparticle is observed at 0.3 μm, with values exponentially decreasing as the wavelength increases. On the other hand, TiO${}_{2}$ has the highest scattering efficiency among these candidates, with the maximum value appearing at 0.45 μm. Similar to Al${}_{2}$O${}_{3}$, the scattering efficiency of the TiO${}_{2}$ microparticle decreases with the increasing wavelength, but its value is much larger than that of Al${}_{2}$O${}_{3}$. Additionally, TiO${}_{2}$ microparticles, with high scattering efficiency below 1.5 μm, are most suitable for achieving a high cooling performance, as 90% of solar irradiation is concentrated in this range. Therefore, we selected the TiO${}_{2}$ microparticle for high scattering in the solar spectrum, despite the occurrence of slight absorption by these microparticles under 0.39 μm.

In the next step, detailed parameters such as the microparticle diameter, microparticle volume fraction, and film thickness were designed through optical analysis. According to previous studies, it is evident that the spectral optical properties, such as reflectance and emittance, of particle-embedded structures change depending on their parameters [49]. To minimize failures in the design and fabrication of the radiative cooling film, we conducted parametric studies on the optical properties (spectral reflectance and emittance) of films using the Monte Carlo method. The Monte Carlo method is a statistical analysis method used to predict possible events with uncertainty [15,39]. In this study, we employed this method to predict the spectral reflectance and spectral emittance of films for radiative cooling.

As mentioned above, the calculations using the Monte Carlo method provide spectral reflectance and emittance. However, radiative cooling is influenced not only by these optical properties but also by factors such as solar irradiation and target temperature. Therefore, a performance indicator that comprehensively considers these factors is commonly adopted. The average reflectance in the solar spectrum and the average emittance in the long-wave infrared region serve as these indicators and are defined as follows [50,51]:(3)$${\bar{R}}_{solar}=\frac{\int_{0.3\mu m}^{2.5\mu m}I_{AM1.5}\left(\lambda \right)R\left(\lambda \right)d\lambda }{\int_{0.3\mu m}^{2.5\mu m}I_{AM1.5}\left(\lambda \right)d\lambda }$$
(4)$${\bar{\varepsilon }}_{LWIR}=\frac{\int_{8\mu m}^{13\mu m}I_{BB}\left(\lambda \right)\varepsilon \left(\lambda \right)d\lambda }{\int_{8\mu m}^{13\mu m}I_{BB}\left(\lambda \right)d\lambda }$$
where R(λ) and ε(λ) are the spectral reflectance in the solar spectrum and spectral emittance in the LWIR (long-wave infrared) atmospheric window, respectively. $I_{AM1.5}\left(\lambda \right)$ is the AM1.5 solar irradiance spectrum. $I_{BB}=\left(\frac{2hc^{2}}{\lambda^{5}}\right)/\left[e^{kc/\lambda k_{B}T}-1\right]$ is the blackbody emission spectral intensity where *T*, *h*, *c*, and $k_{B}$ are the temperature of the film, Planck constants, speed of light, and Boltzmann constants, respectively.

Figure 2a shows the average reflectance (${\bar{R}}_{solar}$) and average emittance (${\bar{\varepsilon }}_{LWIR}$) depending on the TiO${}_{2}$ microparticle diameter. In these calculations, we assumed a constant microparticle volume fraction ($f_{v}$) of 0.3 and a constant film thickness (*t*) of 2 mm to observe how the average reflectance and emittance change with the microparticle diameter. This assumption is reasonable as our design objective is to fabricate a thin structure, and the empirical limit of the microparticle volume fraction is 0.7 [52]. The spectral reflectance in the solar spectrum and spectral emittance in the atmospheric window are included in Appendix A.

To achieve an ideal radiative cooling film, the average reflectance and emittance should approach 1. From an overall perspective, these indicators consistently show high values around 0.9, irrespective of the microparticle diameter. Considering this, it is clear that the material selection of the medium and microparticles is appropriate. In particular, the average emittance maintains a constant value of 0.95 regardless of the microparticle diameter and volume fraction, as shown in Figure 2. This positive effect is attributed to the intrinsic optical properties of PDMS. In contrast, the average reflectance in the solar spectrum varies between 0.89 and 0.937 depending on the microparticle diameter. In particular, the average reflectance is highest at 0.937 for 0.3 μm. This observation can be interpreted as an effect of the scattering efficiency of an individual TiO${}_{2}$ microparticle, which depends on its diameter. Consequently, the TiO${}_{2}$ microparticle diameter is determined to be 0.3 μm [53].

In the final step, we conducted additional calculations to determine the microparticle volume fraction under the specified 2 mm thickness conditions. It is a general phenomenon that an increase in the microparticle volume fraction leads to an enhanced cooling performance, as presented in Figure 2b. However, as the microparticle volume fraction increases, the benefits of flexible radiative cooling, such as flexibility, productivity, and durability, simultaneously decrease. Therefore, we theoretically examined the enhancement in the average reflectance and emittance depending on the microparticle volume fraction and set a challenge to fabricate as high a volume fraction as possible ($f_{v}$ = 0.3).

### 2.3. Fabrication

The fabrication method of FRC film is illustrated in Figure 3a. In the fabrication of the FRC film with PDMS and TiO${}_{2}$ microparticles, achieving a uniform dispersion of microparticles within the PDMS is crucial. This is because the utilization of a non-uniformly dispersed mixture in the fabrication process can lead to non-uniform cooling performance and a lack of durability due to microparticle agglomeration. Previous studies have proposed various approaches to address this challenge, including the use of low viscosity fluids such as toluene and acetone. Lin et al. [43] used a low viscosity toluene solvent to lower the viscosity of PDMS and disperse microparticles easily. However, these solvents pose a critical issue, such as causing nervous system dysfunction in the human body [54]. Therefore, we aim to develop a fabrication method that is harmless to the human body and makes it easier to disperse microparticles in PDMS.

Silicone oil with a low viscosity (μ = 1 cSt) is employed in the mixing process of the PDMS and TiO${}_{2}$ microparticles. As well as silicone oil having low viscosity, it is also well-dispersed with PDMS because of its non-polar characteristic. Additionally, silicone oil is a harmless solvent often used in the food and pharmacy fields. Thus, we decided to employ silicone oil as the buffer medium in our fabrication process.

The FRC film is fabricated through six steps, as shown in Figure 3a. (1) All components, including the TiO${}_{2}$ microparticles, PDMS, and silicone oil, are combined. In this process, the weight ratio of the PDMS, curing agent, and silicone oil is consistently maintained at 10:1:5 (with a volume ratio of 10:1.1:6.9), except for the TiO${}_{2}$ microparticles. (2) The entire composition is roughly mixed for 30 min using a vortex mixer (Scientific Industries, Bohemia, NY, USA, Vortex-Genie 2). (3) The ultrasonicator (Sonictopia, Cheongju, Republic of Korea, STH-750S) is used to thoroughly disperse microparticles under the following conditions: 37.5 W power intensity, 20 KHz frequency, and 100 cycles. After ultrasonication, a well-dispersed white mixture is obtained. (4) The white mixture is poured into a dish to shape the film. (5) The poured white mixture is degassed for 30 min in the vacuum chamber. This step is essential for fabricating a smooth surface film because dissolved gases usually create sinkholes in the medium. (6) The degassed white mixture is heated at 55 °C for 12 h.

As illustrated in Figure 3b, the successful production of a flexible radiative cooling film is achieved after the entire fabrication process. The fabricated FRC film exhibits excellent flexibility and a smooth surface through our new fabrication process despite TiO${}_{2}$ microparticles being added within PDMS. In addition, our fabrication method addresses the non-uniform dispersion of microparticles with a simple and non-toxic fabrication process compared to conventional radiative cooling technologies. Consequently, our simple, inexpensive, and safe fabrication process facilitates scalability and extends the possibility of its practical application for radiative cooling technology.

### 2.4. Experimental Method

To evaluate the performance of the fabricated radiative cooling film, we conducted measurements of the spectral reflectance and spectral emittance using UV-Vis spectroscopy (Shimazu, Kyoto, Japan, UV-3600i plus) and Fourier-transform Infrared spectroscopy (Thermo Fisher Scientific, Waltham, MA, USA, Nicolet iS50). Consistent with the calculation conditions, we positioned an aluminum substrate (*t* = 1 mm) beneath the film during the measurement.

We measured the actual diameter of the TiO${}_{2}$ microparticles to compare with the designed microparticle diameter. First, images of the TiO${}_{2}$ microparticles were taken with a scanning electron microscope (Tescan, Brno, Czech Republic, MIRA3 LMH). After that, we averaged the width and length of the TiO${}_{2}$ microparticles. The width is the longest dimension of the microparticle measured at right angles to the length, and length means the most extended dimension from edge to edge. The length and width are most commonly used for measuring the microparticle diameter [55].

## 3. Results and Discussion

### 3.1. Optical Properties

Figure 4 shows the measured reflectance and emittance with the calculation results of the FRC film, respectively. In general, the spectral reflectance of both films decreases steeply in the ultraviolet regime. As mentioned and shown in Figure 1b above, the absorption efficiency of the TiO${}_{2}$ microparticle is high in the ultraviolet regime (maximum at 0.41 μm). The high absorption efficiency is induced by the high value of the imaginary part of the refractive index, which is the intrinsic optical property of TiO${}_{2}$ microparticles [2,56]. On the other hand, the calculated and measured results exhibit high spectral reflectance in the visible regime. In detail, the average reflectance of the FRC film in the visible regime (0.4–0.8 μm) is 0.98 from the measurement result. This high spectral reflectance from the scattering effect is caused by the significant difference in the refractive index between the TiO${}_{2}$ microparticle and PDMS. In the near-infrared regime, the measured spectral reflectance of the films is relatively lower than in the visible regime. Moreover, several absorption peaks appear at 1.6–1.8 μm and 2.1–2.4 μm. In Figure 5b, the scattering peak’s shift toward the long wavelength regime is observed as its diameter increases. From this point of view, it is possible to analyze whether the scattering efficiency of the single microparticle affects the entire reflectance.

The difference between the calculated and the measured results are analyzed from the perspectives of the microparticle diameter, microparticle volume fraction, and film thickness. To begin with, the TiO${}_{2}$ microparticle diameter exhibiting the optimal cooling performance was determined to be 0.3 μm through the preceding design process. However, it is practically impossible to produce TiO${}_{2}$ microparticles with a perfect spherical shape and uniform size distribution through chemical synthesis methods [57,58]. Thus, we utilized microparticles with a distribution ranging from 0.2 to 0.4 μm, as described above, and Figure 5a shows the size distribution of them. According to Figure 5a, TiO${}_{2}$ microparticles have a size distribution from 0.065 μm to 0.482 μm, with an average diameter of 0.253 μm. Compared to the target diameter, 0.047 μm, smaller TiO${}_{2}$ microparticles were employed. To analyze the effects of this difference, calculations of the absorption/scattering efficiencies of TiO${}_{2}$ single microparticles were performed. Figure 5b shows the efficiencies with respect to wavelength. The scattering efficiency of TiO${}_{2}$ microparticles shifts toward the short wavelength region as the microparticle diameter decreases, and the scattering efficiency in the range of 1 μm to 2.5 μm decreases simultaneously. Conversely, the absorption efficiency observed in the range of 0.3 μm to 0.4 μm is not sensitive to the microparticle diameter. Because of the smaller microparticles, the decrease in the scattering efficiency from 1 μm to 2.5 μm consequently leads to the difference in spectral reflectance in the same region, as represented in Figure 4a and Appendix A.

The microparticle volume fraction is also a suspicious factor for the difference. As discussed earlier, the higher microparticle volume fraction leads to a higher cooling performance. In this study, we aimed to disperse TiO${}_{2}$ microparticles as much as possible by adding low viscosity silicone oil to the high viscosity PDMS. In the end, this study accomplished the maximum volume fraction of 0.331 (4 g) following the fabrication process in Figure 3a. However, it is inevitable to lose microparticles during the process because the mixture should be transferred for dispersion and curing. As a result, the microparticle volume fraction in the fabricated film is apparently lower than the target value. With this observation and Appendix A, it is possible to explain the difference in the calculation and measurement. For instance, if the microparticle volume fraction decreases from 0.3 to 0.1, the spectral reflectance in the solar spectrum does not change significantly, but it decreases significantly in the wavelength range above 1.5 μm. Therefore, it is possible to deduce that the microparticle loss made uncertainties since the difference in the identical range is distinct.

The last perspective for analysis is film thickness. The thickness of the film fabricated in this study is 1.5 mm, which is 0.5 mm thinner than the targeted 2 mm in the design. The difference in film thickness is easily caused by the shrinkage during the curing process and the capillary phenomenon between the mixture and container walls [59]. However, when the film thickness exceeds 500 μm, the spectral reflectance change is negligible, as shown in Appendix A. Therefore, it is unreasonable to analyze whether the difference in spectral reflectance is induced by the film thickness error.

### 3.2. Cooling Power

The cooling power, considering all the heat exchange processes simultaneously, is a performance index used to evaluate the cooling performance of radiative cooling. In detail, the cooling power ($P_{net}$) is the sum of the thermal radiation to space ($P_{rad}$),the absorbed energy from the sun ($P_{sun}$) and the atmosphere ($P_{atm}$), and the transferred energy through conduction and convection ($P_{nonrad}$).

Figure 6 presents the cooling power of the calculated and fabricated FRC films depending on the temperature difference. A positive value of $P_{net}$ indicates the capability of the FRC film to cool a target below the ambient temperature [5]. When the temperature difference is zero, the cooling power of the calculated and fabricated FRC films is 81.7 W/m${}^{2}$ and 67.1 W/m${}^{2}$, respectively. As discussed above, this difference (14.6 W/m${}^{2}$) is caused by decreased spectral reflectance in the fabricated FRC film. Conversely, it is possible to estimate the surface temperature by assuming the cooling power is zero. Figure 6 also indicates that the fabricated FRC film can cool down the surface by up to 6.7 °C compared to the ambient temperature. Comparing the cooling performance of state-of-the-art radiative cooling materials, which achieved approximately 40–100 W/m${}^{2}$ [30,60,61,62], the FRC film is comparable. In addition, our FRC film can cool targets at a constant heat flux of 67.1 W/m${}^{2}$ regardless of their curvature due to its excellent flexibility, safety, and durability.

Contrary to spectral reflectance, the spectral emittance of the FRC film corresponding to the atmospheric window is not significantly changed depending on the microparticle diameter, microparticle volume fraction, and film thickness. In other words, the spectral emittance of the FRC film is robust for the parameters. Hence, it is deduced that the difference between the calculated and measured results shown in Figure 6 is primarily caused by the spectral reflectance in the solar spectrum. Therefore, this result implies that it is necessary to achieve high scattering efficiency microparticles and high microparticle volume fraction fabrication as much as possible.

### 3.3. Ultrasonication Effect

According to previous studies, microparticles are well known for easily forming agglomerations [63]. Due to their Van der Waals force and Brownian random motion, microparticles tend to cluster and bind together. To address the agglomeration of TiO${}_{2}$ microparticles and achieve uniform dispersion, this study employed an ultrasonication process, as shown in Figure 3a. Furthermore, we quantitatively investigated the effect of the ultrasonication process on the cooling performance of FRC films.

Figure 7a shows the spectral reflectance in the solar spectrum depending on the ultrasonication. The ‘w/o’ (red line) indicates the fabricated FRC film without the ultrasonication treatment, and ‘w/’ (blue line) indicates the treated case. Both films exhibit a similar tendency in the ultraviolet regime. However, the FRC film fabricated without ultrasonication treatment shows a slightly lower reflectance in the visible and near-infrared regimes compared to the treated case. It is not a large difference, but it leads to a 7 W/m${}^{2}$ decrease in the cooling power as presented in Figure 7b. Although this value looks small, it is equivalent to about 10.4% of the cooling power, so it is not a negligible effect.

Specifically, this result can be discussed with regard to the scattering efficiency change due to microparticle agglomeration. Contrary to a single microparticle, they show different optical characteristics when a few microparticles agglomerate in the mixture. This agglomeration situation can be simply approximated by the increase in the microparticle diameter. When the microparticle diameter increases, the scattering efficiency spectrum is shifted toward the long wavelength range. As a result, this change brings out the enhancement of spectral reflectance in the near-infrared regime, so it has a negative effect on radiative cooling, where the reflection of solar energy is crucial. Similar to our discussion, Wang et al. [64] reported that the spectral reflectance in the solar spectrum decreases as many microparticles agglomerate. Considering the measured result and the previous studies, it can be concluded that ultrasonication treatment is essential to fabricating high-performance radiative cooling film based on microparticles.

## 4. Conclusions

The present study proposes the structure and fabrication process of a flexible radiative cooling film with high cooling performance. In order to design the flexible radiative cooling film, the composition of the materials, the microparticle diameter, microparticle volume fraction, and film thickness were determined. The TiO${}_{2}$ microparticle, with high scattering efficiency, was selected through Mie scattering theory, and its diameter, volume fraction, and film thickness were determined as 0.3 μm, 0.3, and 2 mm, respectively, by using the Monte Carlo method. In addition, we proposed a novel fabrication process that employs low viscosity silicone oil to disperse TiO${}_{2}$ microparticles evenly in high viscosity PDMS. Thus, the fabricated FRC film achieved a cooling power of 67.1 W/m${}^{2}$ and a cooling temperature of −6.7 °C, with an average reflectance of 0.92 in the solar spectrum and an average emittance of 0.96 in the atmospheric window. Furthermore, the differences between the calculated and measured cooling performances of the fabricated FRC film were analyzed in terms of the microparticle diameter, size distribution, and microparticle volume fraction. From the analyses, we concluded that the TiO${}_{2}$ microparticles were 0.047 μm smaller than intended, the microparticles’ size distribution ranged from 0.065 μm to 0.482 μm, and the inevitable microparticle loss during the dispersion process resulted in a 14.6 W/m${}^{2}$ difference in cooling performance. Finally, the effect of ultrasonication during the film fabrication process was quantitatively discussed. In conclusion, the proposed TiO${}_{2}$-microparticle-based radiative cooling film satisfied the requirement of having a high cooling performance, flexibility, and safety in the fabrication process simultaneously. Hence, the FRC film holds immense potential for future applications in various fields, including architecture, machinery, and energy.

## Figures and Tables

**Figure 1 micromachines-14-02223-f001:**
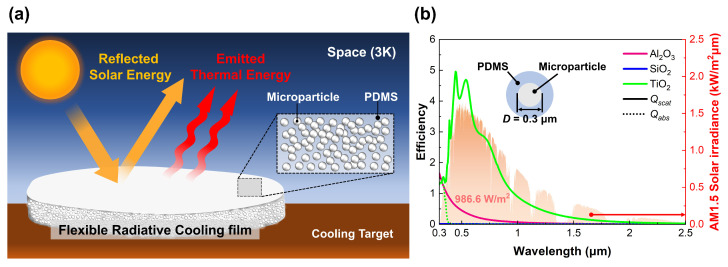
Design and material selection of FRC film. (**a**) Schematic of FRC film. (**b**) Scattering and absorption efficiencies of Al${}_{2}$O${}_{3}$, SiO${}_{2}$, TiO${}_{2}$ microparticles.

**Figure 2 micromachines-14-02223-f002:**
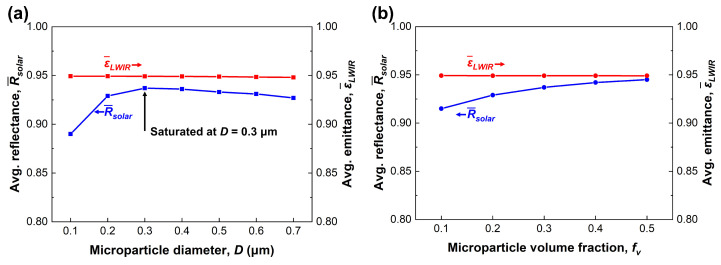
Performance indices (average reflectance and emittance) depending on: (**a**) microparticle diameter (**b**); microparticle volume fraction.

**Figure 3 micromachines-14-02223-f003:**
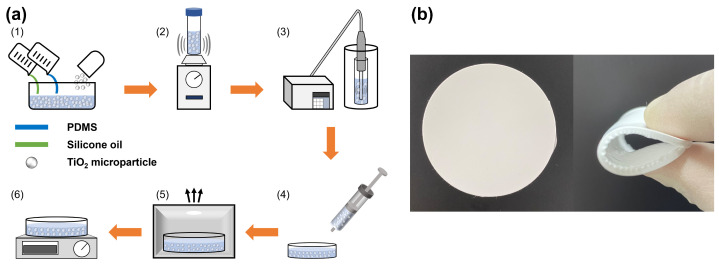
Fabrication detail of FRC film. (**a**) Process. (**b**) Product.

**Figure 4 micromachines-14-02223-f004:**
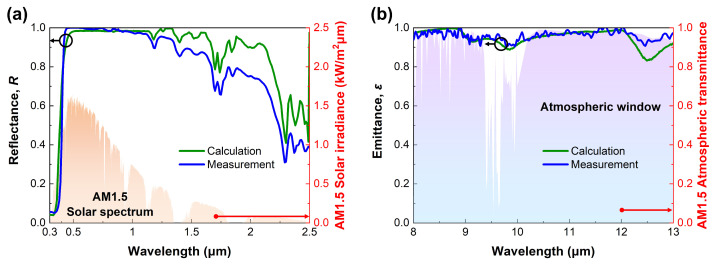
Calculated and measured optical properties of the FRC film in (**a**) solar spectrum and (**b**) atmospheric window.

**Figure 5 micromachines-14-02223-f005:**
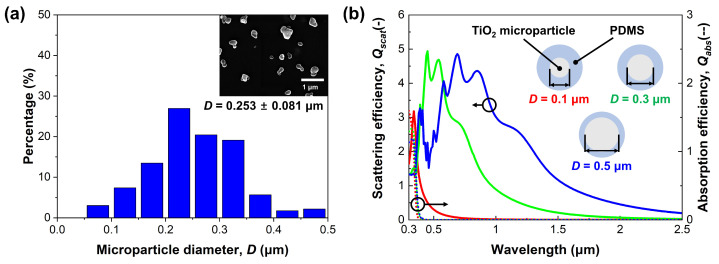
Analysis of TiO${}_{2}$ microparticles. (**a**) Size distribution. (**b**) Scattering (solid line) and absorption (dotted line) efficiencies depending on diameter.

**Figure 6 micromachines-14-02223-f006:**
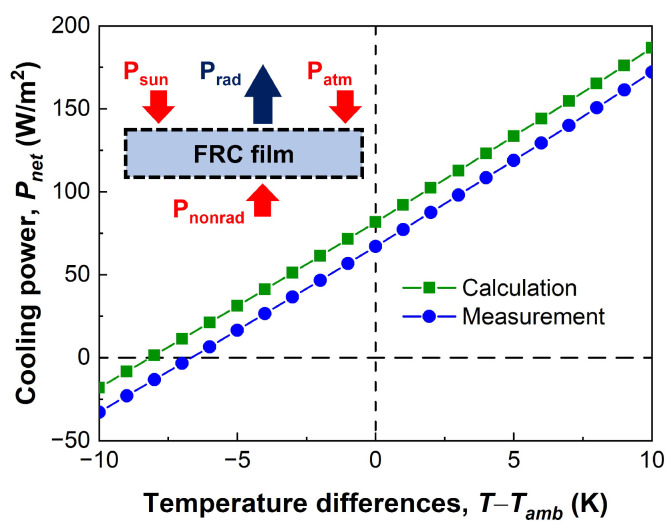
Cooling power of the FRC film depending on the temperature difference.

**Figure 7 micromachines-14-02223-f007:**
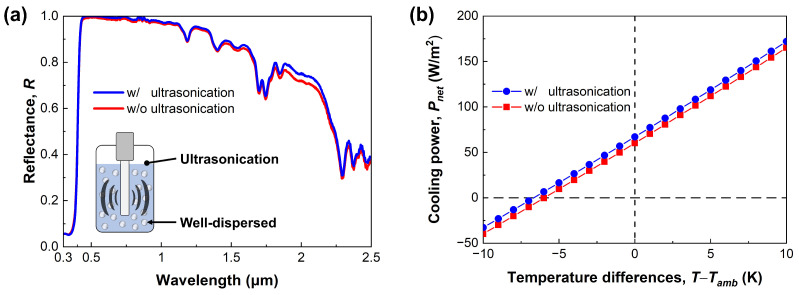
Ultrasonication effect on the FRC films. (**a**) Measured spectral reflectance in the solar spectrum. (**b**) Cooling power.

## Data Availability

Data are contained within the article and Appendix A.

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
