# Peer review of "Development of High-Performance Flexible Radiative Cooling Film Using PDMS/TiO2 Microparticles"

_micromachines, 2023, doi:10.3390/mi14122223_

Round 1

Reviewer 1 Report

Comments and Suggestions for Authors

1. Section 2.2. Design is worth rewriting as it contains both elements of the problem statement, description of theoretical methods and results. Please keep here only the description of theoretical methods and put the rest in the appropriate sections.

2. In the materials and methods section it is necessary to move or add a description of the experimental methods of research: electron microscopy, UV-visible spectroscopy and cooling power measurements.

3. The paper lacks a description of the conditions of ultrasonic experiments: ultrasonic generator model, duration and power of ultrasonic treatment, temperature. In the process of ultrasonic processing there is often contamination of metal titanium from the tip, how could this affect the properties of the films?

4. In the paper, it is worthwhile to distinguish more clearly between the results of theoretical and experimental methods, e.g. signing the figures as "Calculated reflectance spectra..." etc.

5. How the phase composition and crystallite size of titanium dioxide affects the optical and cooling power properties? Please specify the phase composition and crystallite size for the titanium dioxide used in the paper.

Author Response

The authors present a flexible radiative cooling film based on TiO2 particles on PDMS substrates. The film design was based on meticulous theoretical calculations. A systematic investigation was also done to explain the difference between experiments and calculations. I listed a couple of comments for authors to address before it can get published.

  1. Line 30 - Why flexibility is considered a limitation for practical application needs to be justified. For example, since these are used for building applications, the devices do not have to be flexible to be outside of buildings. The advantages need solid justifications. Also, can the particle-based structure proposed in this paper be considered to have good scalability? Why? References are needed to support the arguments.

[AR]: We considered flexibility as one of the limitations in practical applications for radiative cooling techniques. The reason is that cooling targets, including buildings and vehicles, have complex surfaces, such as curved surfaces, rather than flat surfaces. For example, radiative cooling devices, which include Ag or Au metal substrates, are difficult to attach to the curved surface due to lack of flexibility. However, a flexible radiative cooling film can cool them regardless of the curvature of the targets.

Also, scalability is a crucial requirement for the practical application of radiative cooling techniques. While photonic structures face challenges in meeting large-scale production demands due to complicated fabrication processes and expensive materials, particle-based structures demonstrate excellent scalability. Prior research reported that particle-based radiative cooling films are more likely to be commercialized due to their cost-effectiveness and simplicity in fabrication as well as high solar reflectance and thermal emission Ref [38](Yu et al., 2021). In our work, the flexible radiative cooling(FRC) film is composed of PDMS and TiO2 microparticles without a metallic reflecting layer, ensuring a low-cost and straightforward fabrication process compared to the photonic structure.

For clarification, we added the sentence in line 31 of the revision: “Flexibility, scalability, and simple fabrication process are crucial for applying radiative cooling techniques, but photonic structures still pose challenges in satisfying these requirements Ref [7,27](Wang et al., 2020, Xie et al., 2023).” We modified the sentence in line 40 of the revision: “Due to their simple compositions, the particle-based structures are easier to fabricate compared to photonic structures, ensuring excellent flexibility and scalability Ref [37,38](Ding et al., 2022, Yu et al., 2021).”

[Figure AR1]: Performance indices(average reflectance and average emittance) depending on (a) Microparticle diameter (b) Microparticle volume fraction.

  1. Line 97 – references are needed to support the argument that PDMS has the highest spectral emittance compared to other flexible polymers.

[AR]: We chose PDMS as the polymer matrix for its excellent flexibility and high emittance in the atmospheric window, as the polymer matrix. Prior research reported that PDMS exhibits relatively higher absorption in the mid-infrared regime than polymer materials such as polycarbonate(PC), polyethylene terephthalate(PET), polymethylmethacrylate(PMMA), and polyvinylidene fluoride(PVDF) Ref [46](Chen et al., 2021). Due to high absorptance(=emittance) in the mid-infrared regime, many radiative cooling researchers have used PDMS as a polymer matrix in their fabrications.

To clarify our description, the sentence added in line 98 of the revision: “The polymer satisfying our design objectives is PDMS, which has excellent flexibility and high thermal emission. It exhibits higher absorptance(=emittance) in the mid-infrared regime compared to other polymer matrix such as polycarbonate(PC), polyethylene terephthalate(PET), polymethylmethacrylate(PMMA), and polyvinylidene fluoride(PVDF) Ref [46](Chen et al., 2021).”

  1. Figure 2 - arrows should be used to indicate which data is which in a double Y-axis plot.

[AR]: We added arrows with corresponding colors to distinguish data in a double Y-axis plot. To clarify our description, Figure.2 on page 4 of the revision is modified as shown in Figure. AR1.

  1. Figure 3 - A description of each process in Figure 3 (a) is required for clarity. For Figure (b) - it seems unnecessary to show an image that indicates PDMS-made film can be bent - everyone knows PDMS is flexible. No new information is actually provided by showing the images. Authors should consider deleting it or giving some images with labeling that are actually informative to readers.

[AR]: We concede the reviewer's concern about the difficulty in understanding each step of the fabrication process. In addition, PDMS is a general material that has excellent flexibility. However, unlike the bare PDMS film without any microparticles, our FRC film is a composite material of high concentration of TiO2 microparticles and PDMS. This means that their compositions are different, affecting their mechanical properties. Therefore, we would like to demonstrate that our FRC film maintains excellent flexibility even when microparticles in PDMS are embedded.

For clarification, the original sentences were modified in line 201 - 212 of the revision: “The FRC film is fabricated through six steps, as shown in Figure. 3(a). (1) All components, including TiO2 microparticles, PDMS, and silicone oil, are combined. In this process, the weight ratio of PDMS, curing agent, and silicone oil is consistently maintained at 10:1:5 (with a volume ratio of 10:1.1:6.9), except for TiO2 microparticles. (2) The entire composition is roughly mixed for 30 minutes using a vortex mixer(Scientific Industries, Vortex-Genie 2). (3) The ultrasonicator(Sonictopia, STH-750S) is used to thoroughly disperse microparticles under the following conditions: 37.5 W power intensity, 20 KHz frequency, and 100 cycles. After ultrasonication, a well-dispersed white mixture is obtained. (3) The white mixture is poured into a dish to shape the film. (4) The poured white mixture degassed for 30 minutes in the vacuum chamber. This step is essential for fabricating a smooth surface film because dissolved gases usually create sinkholes in the medium. (6) The degassed white mixture is heated at 55℃ for 12 hours.”

  1. Line 188 - 192 - it seems to be a novel method to mix TiO2 and PDMS in silicone oil. Authors can emphasize the originality of this process by comparing it to the existing similar processes in the literature by providing some references.

[AR]: We appreciate the reviewer for recognizing the novelty of our fabrication method, which involves mixing TiO2 microparticles and PDMS in low viscous silicone oil. Adding low viscous silicone oil is harmless to humans and has the advantage of uniformly dispersing TiO2 microparticles in high viscous PDMS. Unlike particle-based radiative cooling techniques using toluene or acetone as a buffer medium, our method addresses the toxic issues of these solvents and uniformly disperses microparticles.

For clarification, we added a sentence in line 191 of the revision: “Ref [43](Lin et al., 2022) used low viscous toluene solvent to lower the viscosity of PDMS and disperse microparticles easily.”

  1. Line 204 - The authors claim in the introduction that their method is scalable and easy to do. Please comment on such feasibility using the said method, and compare it to several existing technologies.

[AR]: As aforementioned, scalability is a crucial requirement for the practical application of radiative cooling techniques. Various approaches, including photonic and particle-based structures, have been investigated. Firstly, Ref [60](Raman et al., 2014) serves as a representative study of photonic structures, producing a multilayer structure made of HfO2, SiO2, Ti, Ag, and Si using physical vapor deposition(PVD). This well-known manufacturing process is still challenging to commercialize due to the sophistication and expense of the processes and the high cost of materials.

In contrast, particle-based radiative cooling structures have the advantage of low cost of materials and a simple fabrication process, making them easier to commercialize compared to photonic structures. Despite these advantages, particle-based radiative cooling structures have non-uniform microparticle dispersion problems within the polymer matrix. Accordingly, toluene or acetone has been used in prior research, but these solvents are toxic to humans. To address this problem, we utilized a low viscous silicone oil solvent, which is harmless to humans, to disperse microparticles uniformly and proceed with the third process, ‘ultrasonication’ in Figure. 3a. Therefore, we fabricated the flexible radiative cooling film with our new fabrication method, which is excellent for a simple fabrication process, cost-effectiveness, and non-toxic aspects. Ultimately, we can address concerns about the feasibility of large-scale production and practical applications of radiative cooling by achieving high cooling performance, flexibility, and safety.

To clarify the description, the sentences were added in line 214 of the revision: “The fabricated FRC film exhibits excellent flexibility and a smooth surface through our new fabrication process despite TiO2 microparticles being added within PDMS. In addition, our fabrication method addresses the non-uniform dispersion of microparticles with a simple and non-toxic fabrication process compared to conventional radiative cooling technologies. Consequently, our simple, inexpensive, and safe fabrication process facilitates scalability and extends the possibility of practical application for radiative cooling technology.”

  1. Line 274 to 281 - the achieved cooling power should be compared to start-of-the-art values and discuss the advantages and disadvantages of the proposed technology.

[AR]: We agree with the reviewer's comment that the cooling power we achieved with our fabricated FRC film should be compared with existing radiative cooling technologies. We compared two types of radiative cooling research, photonic and paint-type particle-based structures. Firstly, Ref [60](Raman et al., 2014) mentioned in the previous question is a representative study of photonic structure and the first case of daytime radiative cooling. In their study, various materials, including HfO2, SiO2, Ti, Ag, and Si, were fabricated into a multilayer, achieving a cooling power of 40.1 W/m2. However, this technology requires a sophisticated fabrication process and a high cost of materials. Therefore, our FRC film has many advantages, such as high cooling power, simple fabrication methods, and low cost of materials compared to them.

Next is a paint-type particle-based radiative cooling technology. In Ref [62](Chae et al., 2021), Al2O3 and SiO2 microparticles, DPHA, and IRGACURE 184 were used to fabricate the paint layer, and this paint achieved approximately cooling power of 100 W/m2. Despite this paint technology having a cooling power of 33 W/m2 higher than ours, it has a problem with weak durability. Radiative cooling material should reflect the incident solar energy by attaching to the exterior of the cooling target. However, the paint causes cracks when exposed to the outside for a long time due to weak durability. This indicates that the cooling power can decrease by the crack, and it causes less reliability in the cooling performance. Eventually, our film has lower cooling power but excellent flexibility, safety, and durability. In practical applications, more uniform cooling performance can be implemented.

For clarification, we added the sentences in line 306 of the revision: “Comparing the cooling performance of state-of-the-art radiative cooling materials, which achieved approximately 40 - 100 W/m2 Ref [30,60-62](Altamimi et al., 2023, Raman et al., 2014, Jiang et al., 2023, Chae et al., 2021), the FRC film is comparable. In addition, our FRC film can cool targets at a constant heat flux of 67.1 W/m2 regardless of their curvature due to excellent flexibility, safety, and durability.”

Again, we thank all the reviewers for their valuable time. We sincerely hope that this response letter clarifies the reviewers' concerns.

Reviewer 2 Report

Comments and Suggestions for Authors

The authors present a flexible radiative cooling film based on TiO2 particles on PDMS substrates. The film design was based on meticulous theoretical calculations. A systematic investigation was also done to explain the difference between experiments and calculations. I listed a couple of comments for authors to address before it can get published.

1. Line 30 - Why flexibility is considered a limitation for practical application needs to be justified. For example, since these are used for building applications, the devices do not have to be flexible to be outside of buildings. The advantages need solid justifications. Also, can the particle-based structure proposed in this paper be considered to have good scalability? Why? References are needed to support the arguments.

2. Line 97 - references are needed to support the argument that PDMS has the highest spectral emittance compared to other flexible polymers. 

3. Figure 2 - arrows should be used to indicate which data is which in a double Y-axis plot.

4. Figure 3 - A description of each process in Figure 3 (a) is required for clarity. For Figure (b) - it seems unnecessary to show an image that indicates PDMS-made film can be bent - everyone knows PDMS is flexible. No new information is actually provided by showing the images. Authors should consider deleting it or giving some images with labeling that are actually informative to readers.

5. Line 188 - 192 - it seems to be a novel method to mix TiO2 and PDMS in silicone oil. Authors can emphasize the originality of this process by comparing it to the existing similar processes in the literature by providing some references.

6. Line 204 - The authors claim in the introduction that their method is scalable and easy to do. Please comment on such feasibility using the said method, and compare it to several existing technologies.

7. Line 274 to 281 - the achieved cooling power should be compared to start-of-the-art values and discuss the advantages and disadvantages of the proposed technology.

Comments on the Quality of English Language

See several minor comments related to Figures and their captions.

Author Response

[REVIEWR2]

  1. Section 2.2. Design is worth rewriting as it contains both elements of the problem statement, description of theoretical methods and results. Please keep here only the description of theoretical methods and put the rest in the appropriate sections.

[AR]: We found your comment to be reasonable and discussed how to better represent our research. Given that this paper aims to propose both material compositions and the entire fabrication process, including design, fabrication, and measurement, we concluded that incorporating the analysis of theoretical design into Section 2.2 would enhance its effectiveness. Moreover, we thought that providing detailed information about our determination of fabrication factors in this section is appropriate. Thus, despite appreciating your opinion, we have decided to maintain the current configurations.

  1. In the materials and methods section it is necessary to move or add a description of the experimental methods of research: electron microscopy, UV-visible spectroscopy and cooling power measurements.

[AR]: We agree with the comment of the reviewer and added a new section for experimental methods about UV-Vis and FTIR spectroscopy, and a scanning electron microscope(SEM).

For clarification, we added the new Section 2.4 Experimental method in line 221 of the revision:

2.4 Experimental method

To evaluate the performance of the fabricated reflective cooling film, we conducted measurements of spectral reflectance and spectral emittance using UV-Vis spectroscopy(Shimazu, UV-3600i plus) and Fourier-transform Infrared spectroscopy(Thermo Fisher Scientific, Nicolet iS50). Consistent with the calculation conditions, we positioned an aluminum substrate(t=1 mm) beneath the film during the measurement.

We measured the actual diameter of the TiO2 microparticles to compare with the designed microparticle diameter. First, the images of the TiO2 microparticles were taken with a scanning electron microscope(Tescan, MIRA3 LMH). After that, we averaged the width and length of the TiO2 microparticles. The width is the longest dimension of the particle measured at right angles to the length, and length means the most extended dimension from edge to edge. The length and width are most commonly used for measuring the microparticle diameter Ref [55](CH et al., 2001).

  1. The paper lacks a description of the conditions of ultrasonic experiments: ultrasonic generator model, duration and power of ultrasonic treatment, temperature. In the process of ultrasonic processing there is often contamination of metal titanium from the tip, how could this affect the properties of the films?

[AR]: We agree with the reviewer that our paper needs to improve the description of the ultrasonication process conditions. As mentioned in a detailed description of the fabrication process in Figure. 3(b), the sentence was modified in line 206 of the revision: “(3) The ultrasonicator (Sonictopia, STH-750S) is used to thoroughly disperse microparticles under the following conditions: 37.5 W power intensity, 20 KHz frequency, and 100 cycles.”

As the reviewer mentioned, we are also concerned about the contamination caused by the metal titanium from the tip. We conducted the ultrasonication step with 37.5 W(5%) of the total power intensity 750 W to minimize the contamination caused by the fracture of the tip. Additionally, prior to each ultrasonication step, the tip was clearly washed three times with ethanol to remove impurities. The low power intensity of the ultrasonicator and pre-cleaning have reduced the possibility of contamination due to the metal titanium from the tips. Thus, we can conclude that the FRC film is not affected by the metal titanium from the tip.

  1. In the paper, it is worthwhile to distinguish more clearly between the results of theoretical and experimental methods, e.g. signing the figures as "Calculated reflectance spectra..." etc.

[AR]: We appreciate the reviewer's comments. The terms used for theoretical calculations and experimental results were not clearly distinguished. Consequently, we have designated the theoretically calculated results as 'calculation' and the experimental results as 'measurement.'

To clarify our description, we modified the words of the revision: in the caption of Figure. 4, Optical properties to ‘Calculated and measured optical properties,’ and in line 246 the spectral reflectance of the film to ‘the measured spectral reflectance of the film.’

  1. How the phase composition and crystallite size of titanium dioxide affects the optical and cooling power properties? Please specify the phase composition and crystallite size for the titanium dioxide used in the paper.

 [AR]: Titanium dioxide(TiO2) exists in several phases, most commonly anatase and rutile. In the optical properties of the two phases, the rutile and anatase TiO2 microparticles have 2.74 and 2.57 refractive indexes in wavelengths of 550 nm, respectively. In order to enhance reflectance in the solar

[Figure AR2]: Depending on the diameter of TiO2 microparticles (a) Calculated spectral reflectance in the solar spectrum (b) Cooling power.

spectrum, the refractive index difference between polymer matrix and microparticles must be large. Therefore, we enhance the performance of the FRC film using rutile TiO2 microparticles.

The calculated spectral reflectance and cooling properties according to the size(diameter) of the TiO2 microparticles are shown in Figure. AR2. Depending on the diameter of TiO2 microparticles, the spectral reflectance is different in the ultraviolet, visible, and near-infrared regimes. Through this, it can be described that the diameter of the TiO2 microparticles is a parameter that affects the spectral reflectance in the solar spectrum. Figure. AR2(b) presents the cooling power according to each diameter, and it can be seen that the cooling power depends on the microparticle diameter. Among them, the diameter of 0.3 μm shows the highest cooling power of 81.7 W/m2. Accordingly, the microparticle diameter determined in Section 2.2 Design process was chosen to be 0.3 μm in our work.

For clarification, we added the phase of TiO2 microparticle in line 80 of the revision: “Titanium oxide(TiO2, rutile, >99.5%, 0.2 - 0.4 μm).”

Again, we thank all the reviewers for their valuable time. We sincerely hope that this response letter clarifies the reviewers' concerns.
